# Normalized Pulmonary Artery Diameter Predicts Occurrence of Postpneumonectomy Respiratory Failure, ARDS, and Mortality

**DOI:** 10.3390/cancers12061515

**Published:** 2020-06-10

**Authors:** Elisa Daffrè, Mathilde Prieto, Haihua Huang, Aurélie Janet-Vendroux, Kim Blanc, Yen-Lan N’Guyen, Ludovic Fournel, Marco Alifano

**Affiliations:** 1Department of Thoracic Surgery, Cochin Hospital, AP-HP Center University of Paris, 75014 Paris, France; daffre.elisa@gmail.com (E.D.); mathilde.prieto@wanadoo.fr (M.P.); hhh002559@163.com (H.H.); aurelie.janet-vendroux@wanadoo.fr (A.J.-V.); ludovic.fournel@aphp.fr (L.F.); 2Department of Chest Disease, Cochin Hospital, AP-HP Center University of Paris, 75014 Paris, France; kim.blanc@chi-andre-gregoire.fr; 3Department of Anesthesiology and Intensive Care, Cochin Hospital, AP-HP Center University of Paris, 75014 Paris, France; yen-lan.nguyen@aphp.fr

**Keywords:** pneumonectomy, outcome, pulmonary artery, respiratory failure, ARDS, mortality

## Abstract

Hypothesizing that pulmonary artery diameter is a marker of subclinical pulmonary hypertension, we assessed its impact on postoperative outcome in patients requiring pneumonectomy. Morphometric, clinical, and laboratory data were retrospectively retrieved from files of 294 consecutive patients treated by pneumonectomy for malignancy (289 NSCLC). Pulmonary artery was measured at bifurcation level on CT scan and normalized by body surface area. Median normalized pulmonary artery diameter (cut-off for analyses) was 14 mm/m2. Postoperatively, 46 patients required re-do intubation and 30 had acute respiratory distress syndrome (ARDS). Multivariate analysis showed that Charlson Comorbidity Index >5 (*p* = 0.0009, OR 3.8 [1.76–8.22]), right side of pneumonectomy (*p* = 0.013, OR 2.37 [1.20–4.71]), and higher normalized pulmonary artery diameter (*p* = 0.029, OR 2.16 [1.08–4.33]) were independent predictors of re-do intubation, while Charlson Comorbidity Index >5 (*p* = 0.018, OR 2.55 [1.17–5.59]) and higher normalized pulmonary artery diameter (*p* = 0.028, OR = 2.52 [1.10–5.77]) were independently associated with occurrence of ARDS. Post-operative mortality was 8.5%. Higher normalized pulmonary artery diameter, (*p* = 0.026, OR 3.39 [1.15–9.95]), right side of pneumonectomy (*p* = 0.0074, OR 4.11 [1.46–11.56]), and Charlson Comorbidity Index >5 (*p* = 0.0011, OR 5.56 [1.99–15.54]) were independent predictors of postoperative death. We conclude that pre-operative normalized pulmonary artery diameter predicts the risk of re-do intubation, ARDS and mortality in patients undergoing pneumonectomy for cancer.

## 1. Introduction

Long-term survival of patients with non-small cell lung cancer (NSCLC) remains poor in spite of advances in therapies and refinement in multimodal approach [1]. Operable patients have a better prognosis [1], but a significant percentage of them require pneumonectomy which per se still carries important post-operative morbidity and mortality and modest 5-year survival. In a recently published study on 350 consecutive patients, we showed 30- and 90-day day mortalities of 9%, and 13%, respectively, and 5-year survival of 34% [2], and these results are consistent with previous works of other teams [3,4,5]. This stresses the need to better understand factors associated with short-term outcome, in order to optimize selection of patients for pneumonectomy. Respiratory complications are relatively frequent in pneumonectomy patients and are responsible for a significant percentage of operative mortality: Major respiratory complications occur in 10–20% of cases with progression to acute respiratory distress syndrome (ARDS) in approximately half of them [6,7,8,9]. Patients with post-pneumonectomy ARDS have an extremely high mortality rate with figures ranging 50–90% [6,7,8].

Some studies have addressed risk factors for post-operative respiratory failure and ARDS [6,7,8,10,11,12]: Although the role of intra-operative non-protective ventilation (large tidal volume and increased airway pressures) and fluid administration (non-restriction) has been proved [10,13], patient-related pre-operative factors, including respiratory function parameters, remain matter of debate because even studies with very large sample sizes sometimes failed to find significant correlations [11,14]. A series dealing with non-selected consecutive patients requiring pneumonectomy, showed at univariate analysis a significant association of postoperative forced expiratory volume in 1 s (ppo-FEV_1_) with pulmonary complications, but multivariate analysis failed to prove the independent character of this association [13]. In our recent monocentric study on 543 consecutive patients undergoing pneumonectomy for different indications, mainlyNSCLC, we reported that respiratory failure requiring invasive mechanical ventilation (IMV) and ARDS occurred in 16% and 11% of cases, respectively. At multivariate analysis, only right-side pneumonectomy and higher Charlson Comorbidity Index (CCI) were identified as independent risk factors for acute respiratory distress syndrome, whereas respiratory function parameters did not [15]. 

A historical study showed that high mean pulmonary artery pressure (PAP) measured intraoperatively in patients undergoing pneumonectomy was associated with an increased incidence of cardiorespiratory complications and mortality [16]. Assessment of systolic PAP may be carried out by echocardiography. Although this investigation is not part of recommended routine pre-operative work-up of pneumonectomy patients [17], it is performed liberally in several centers and is mainly used to measure left ventricle ejection fraction. However, to our knowledge, echocardiographic assessment of PAP has not been suggested to predict respiratory complication and mortality of pneumonectomy patients. In lobectomy patients, the presence of high right ventricular systolic pressure measured at echocardiography did not correlate with outcome [18]. 

A previous pilot study of our group on 161 consecutive patients undergoing pneumonectomy showed that normalized pulmonary artery diameter (NPAD) for body surface area (BSA) could be a predictor of postoperative respiratory failure and ARDS [19]. Thus, we aimed at reporting our more mature experience on the topic evaluating, in a larger series of pneumonectomy patients, the impact of several pre-operative parameters, including NPAD and pulmonary artery diameter (PAD)/aorta diameter (AoD) ratio, on the occurrence of post-pneumonectomy respiratory failure and ARDS.

## 2. Results

Between January 2007 and December 2015, 294 patients treated by pneumonectomy for malignancy (289 NSCLC, 3 malignant pleural mesothelioma, 2 pulmonary metastasis from extra-pulmonary primary tumors) in our department had pre-operative thoracic CT scan. The main clinical-pathological characteristics of the study patients are summarized in Table 1. 

### 2.1. Morphometric Measurements 

Results of morphometric measurements are reported in Table 1. Median normalized diameter was 14 mm/m^2^ [13,14,15,16]. Thus, 14 mm/m^2^ was used as cut-off to differentiate the subgroup of patients with increased NPAD.

### 2.2. Correlations between PAD, Morphometric Parameters, Clinical, and Laboratory Variables

Crude PAD was related with BSA (r = 0.315, *p* = 0.000082), providing argument for need of normalization by this parameter. Correlations between NPAD and PAD/AOD ratio with baseline morphometric, clinical, and laboratory variables are shown in Table 2. Higher NPAD was associated with higher age (*p* = 0.0049) and lower BMI (*p* = 0.00024); a trend toward higher PAD was observed in women (*p* = 0.052). Of note, no relation was found between NPAD and respiratory function parameters. 

### 2.3. Factors Associated with Short-Term Outcome: The Role of PAD

Postoperatively, 46 patients required re-do intubation and IMV and 30 of them had ARDS. Relations between patient’s and tumor related factors with occurrence of respiratory failure requiring intubation or ARDS or death are reported in Table 3 and Table 4. 

Need of re-do intubation was related with higher age (*p* = 0.028), right-side of disease (*p* = 0.011), CCI >5 (*p* = 0.00032), and higher NPAD (*p* = 0.010). Multivariate analysis (model including these significant factors) showed that CCI >5 (*p* = 0.0009, OR 3.8 [1.76–8.22]), right side of pneumonectomy (*p* = 0.013, OR 2.37 [1.20–4.71]), and higher NPAD (*p* = 0.029, OR 2.16 [1.08–4.33]) were independent predictors of need of IMV (Table 5).

Univariate analysis showed that transfer coefficient of the lung for carbon monoxide (KCO) was also associated with need of IMV. Because of KCO being assessed in 132 patients, another multivariate analysis model was performed including KCO together with age, side of disease, CCI, and normalized pulmonary artery diameter. In this model significance of KCO was lost whereas CCI, right side of pneumonectomy and NPAD maintained significance. 

Occurrence of ARDS was associated with higher age (*p* = 0.020), right side of pneumonectomy (*p* = 0.043), CCI >5 (*p* = 0.0043) and higher NPAD (*p* = 0.021). Multivariate analyses (model including the above quoted significant factors) showed that CCI >5 (*p* = 0.018, OR 2.55 [1.17–5.59]) and higher NPAD (*p* = 0.028, OR = 2.52 [1.10–5.77]) were independently associated with occurrence of ARDS (Table 5). This was confirmed by another multivariate analysis model built to take into account KCO, significant at univariate (*p* = 0.016) but not at multivariate analysis.

To note, among patients harboring PAD/AoD ratio > 1, the occurrence of postoperative ARDS or IMV was not significantly different compared to other patients (*p* = 1.0 and *p* = 0.37, respectively). 

Furthermore, NPAD predicted postoperative occurrence of re-do intubation (*p* = 0.014) or ARDS (*p* = 0.027) when the 5 patients with malignancy other than NSCLC were excluded.

Last, neither PAP systolic measure taken at echocardiography (*n* = 233) nor estimation of left ventricle ejection fraction predicted the occurrence of re-do intubation or ARDS.

Post-operative mortality was 25/294 (8.5%). Relations between patient and tumor-related factors with mortality are reported in Table 4. At univariate analysis, older age (*p* = 0.00046), right side of pneumonectomy (*p* = 0.0062), lower ppoFEV1 (*p* = 0.047), and CCI >5 (*p* = 0.000086) were significantly associated with postoperative mortality. The association of higher NPAD with postoperative mortality was close to significance at univariate analysis (*p* = 0.06) but reached significance (*p* = 0.026, OR 3.39 [1.15–9.95]) at multivariate analysis (model including age, side, ppoFEV1, CCI, and NPAD), together with right side of pneumonectomy (*p* = 0.0074, OR 4.11 [1.46–11.56]) and CCI >5 (*p* = 0.0011, OR 5.56 [1.99–15.54]) (Table 5).

## 3. Discussion

In the present paper we report evidence that NPAD is a strong and independent predictor of occurrence of post pneumonectomy respiratory failure and ARDS. It also independently predicted post-operative mortality.

In 1962 Rams et al. reported that intraoperative mean PAP was higher in pneumonectomy patients presenting catastrophic post-operative respiratory complications (on the average, 26 mmHg versus 22 mmHg) [16]. Mean PAP measurement is based on right cardiac catheterization, an invasive and possibly morbid procedure which is not recommended in the routine pre-operative work-up of lung cancer possibly necessitating pneumonectomy. Extrapolation of mean PAP from systolic one measured at echocardiography remains matter of debate [20,21]. Anyway, no experience exists in lung cancer patients and, a fortiori, in those requiring pneumonectomy.

In this series we assessed both crude and NPAD and found that normalized one was a much more reliable parameter. Although it has been reported—and confirmed by our findings—that PAD is dependent on BSA [22], normalization of crude measure by this parameter has not been reported so far, in spite of quite common practice of normalizing AoDby BSA in vascular surgery studies [23].

We also assessed PAD/AOD ratio and failed to find any correlation between this parameter and outcome. The PAD/AoD ratio >1 has been reported to indicate pulmonary hypertension [22] and is associated with COPD exacerbation [24]. Probably, this parameter better identifies patent pulmonary hypertension more than subclinical one, which could be detected by increased NPAD. We hypothesize that, as pneumonectomy leads to non-negligible loss of global respiratory function and is associated with significant hemodynamic alteration with increase of the pulmonary blood flow to the remaining lung, in case of pre-existent subclinical pulmonary hypertension, this condition could favor the occurrence of postoperative respiratory failure and ARDS. 

In this study, we identified right side of pneumonectomy as an independent risk factor of respiratory failure and mortality after pneumonectomy. For ARDS, significance at univariate analysis was lost at multivariate analysis. Several studies, including ours in a larger dataset, reported a higher postoperative rate of respiratory failure and ARDS after right-side pneumonectomy [2,9,15,25]. We underlined that this could relate to multiple factors, including the more frequent predominance of the right lung in terms of perfusion and the higher postoperative PAP after right than left pneumonectomy [26]. Preoperative work-up included lung perfusion scan in all patients and echocardiography in almost all. These noninvasive tests are sensitive enough to detect baseline mild pulmonary hypertension and to predict severe postpneumonectomy pulmonary hypertension, which has been reported to be associated with postpneumonectomy ARDS [27]. Apart from preoperative ventilatory function, the adequacy of pulmonary vascular bed and circulation after pneumonectomy may be crucial to predict postoperative respiratory failure and ARDS. Fee and colleagues [28] proposed preoperative assessment of pulmonary vascular resistance by right heart catheterization at rest and during exercise; they suggested that high pulmonary vascular resistance is a better predictor of mortality than simple pulmonary function tests, underlining a possible role of secondary pulmonary hypertension caused by a reduction in the pulmonary capillary bed as a determinant of post-pneumonectomy respiratory failure and ARDS. 

We found that higher CCI (≥5) was independently associated with the three evaluated outcome measures. Comorbidities are known to be associated with a higher risk of postoperative respiratory complications [9]. Postoperative pneumonia is the most frequent complication leading to respiratory failure; of note the occurrence of postpneumonectomy pneumonia is associated with comorbidities, suggesting that comorbidities may predispose to pneumonia, which in turn may evolve toward respiratory failure, ARDS, and death [15,16,17,18,19,20,21,22,23,24,25]. Of note, comorbidities are also associated with a higher risk of postoperative bleeding and need for transfusion, which are well-known risk factors for ARDS.

### Strengths and Limitations of the Study

A high number of consecutive patients undergoing pneumonectomy in a single institution, and whose pre-operative scan was available, were evaluated. To our knowledge assessment of PAD as a predictor of outcome in lung cancer patients requiring pneumonectomy has not been reported so far, except in our pilot study [19]. Main limitation of the present study relies in its retrospective character despite the small number of patients with missing data. In our center, right catherization is not carried out routinely, but discussed on a case-by-case basis in case of patent pulmonary hypertension at echography. On the basis of our results, the idea of performing this invasive assessment of pulmonary pressure in patients with high NPAD (which otherwise reveals moderate increase in pulmonary arterial pressure) should probably be tested in the setting of prospective studies. 

## 4. Materials and Methods

### 4.1. General Study Design

Clinical, laboratory, and pathological data from all consecutive patients who underwent pneumonectomy for malignancy at Paris Center University Hospital (Paris, France) between January 2007 and December 2015 were retrospectively retrieved. Among them, patients whose pre-thoracotomy computed thoracic and upper abdominal scan was available in the Picture Archive and Communication System (PACS) of the hospital were included in the present study. The patients represents a subset of population of previous studies on pneumonectomy outcomes [15,25]. The research was conducted according to recommendations outlined in the Helsinki declaration as well as to French laws on Biomedical Research. An informed consent from all patients and IRB approval was obtained (CERC-SFCTCV-2015-11-4-13-16-16-AlMa/CPPHUPC). 

### 4.2. Pre-Operative Investigations and Peri-Operative Care

In all patients, pre-operative work-up included chest X-ray, fiber-optic bronchoscopy, thoracic and upper abdominal CT scan, as well as contrast-enhanced cerebral CT scan or MRI. PET scan was routinely performed. Neoadjuvant chemotherapy was administered in patients with pathologically confirmed N2-disease NSCLC, who were judged operable only in the presence of clinical response or stable disease. 

Functional assessment included spirometry and perfusion lung scan, allowing calculation of ppoFEV1. Patients were considered operable without further investigation if ppoFEV1 was major to 40% and non-operable if ppoFEV1 was inferior to 30%. In intermediate cases decision of operability was taken on an individual basis taking into account results of DLCO determination and of an exercise test.

Pneumonectomy with radical nodal dissection was carried out through standard thoracotomy in all the cases. All the patients had intraoperative protective ventilation and fluid restriction. All were managed postoperatively in a specialized ICU and returned in the surgical ward on the 4th post-operative day if no complication occurred.

### 4.3. Collected Data

Patients’ characteristics, treatment procedures, and short-term outcomes were collected using a standardized case report form. We recorded: Age; sex; height; weight; medical history, including, tobacco consumption, ischemic heart disease and other co-morbidities; CCI; presence of symptoms; FEV1; FEV1/FVC ratio; ppoFEV1 (calculated on the basis of perfusion scan). We also collected histologic type and tumor stage which was reattributed according to the 7th TNM edition in NSCLC cases [29]. 

We calculated for all the patient’s body surface area according to the standard equation: BSA [m^2^] = weight [kg]^0.425^ × height [cm]^0.725^ × 71.84 [m^2^/Kg × cm].

### 4.4. CT Measurements

Measure of diameters was carried out in a semiautomatic manner by using the measurement tool of the Carestream software (Carestream Health Inc., Rochester, NY, USA). Main PAD and AoD were measured on the same slice at level of pulmonary artery bifurcation, as reported by others [22] and in our previous pilot study [19]. PAD/AoD was then calculated. PAD was considered as crude value and normalized for BSA [19].

### 4.5. Outcomes

Three main outcome measures were evaluated: Need of post-operative IMV (after successful extubation in the recovery room), occurrence of ARDS, and mortality. ARDS was defined as acute onset of hypoxemia, measured as PaO2/FIO2 ratio <300 and infiltrate of the remaining lung on chest radiograph in the absence of clinical and echocardiographic findings of cardiac failure [30]. Early mortality was defined as death occurring within the 30rd postoperative day or during the same hospitalization as pneumonectomy, even if beyond the 30rd postoperative day.

### 4.6. Data Analysis

Data processing and analysis were performed with the statistical Software system SEM (SILEX Developpment, Mirefleurs, France). Results were expressed as percentage, mean ±SD for normally distributed and median (interquartile range) for non-normally distributed quantitative variables. 

Univariate analyses were performed to assess relationships with patients’ characteristics and histological features, and correlations were assessed by the Spearman rank test for continuous variables. Mann–Whitney and Kruskal–Wallis tests were used to perform group comparisons as appropriate. Parameters showing correlations (*p* < 0.1) with different outcome parameters at univariate analysis were entered in a backward logistic regression models to assess their independent value. A *p*-value of less than 0.05 was considered significant.

## 5. Conclusions

Normalized pulmonary artery diameter is a strong predictor of severe respiratory complications after pneumonectomy. In perspective, we could suggest that pulmonary artery measure should be integrated in the risk evaluation of patients requiring pneumonectomy; development of comprehensive prognostic index systems need to be developed in adequately designed studies. 

## Figures and Tables

**Table 1 cancers-12-01515-t001:** Patients′ characteristics.

Features	Total Sample = 294
Age: Years	63.9 ± 9.9
Gender: Men/Women	211 (71.8%)/83(28.2%)
Current/Never Smokers (*n* = 291)	267 (91.8%)/24 (8.2%)
Cumulative tobacco consumption (Pack/Years) (*n* = 291)	40 (30–50)
Right/Left side	135 (45.9%)/159 (54.1%)
Weight (kg)	71.84 ± 14.48
Height (cm)	169.3 ± 8.1
BMI (kg/m^2^)	24.98 ± 4.29; median 24
Underweight <18.5	18 (6.1%)
Normal weight 18.5–25	137 (46.6%)
Overweight 25.01–30	102 (34.7%)
Obesity >30	37 (12.6%)
BSA (m^2)^	1.82 ± 0.19
Diabetes Yes/No (*n* = 291)	24 (8.2%)/267 (91.8%)
Hypertension Yes/No (*n* = 291)	110 (37.8%)/181 (62.2%)
Ischemic Heart Disease Yes/No (*n* = 291)	61 (21.0%)/230 (79.0%)
CCI (*n* = 291)	5.1 ±1.7; median 5.0
Baseline Modified Borg Dyspnea Scale >2 (*n* = 288) Yes/No	53 (18.4%)/235 (81.6%)
FEV1 (% of predicted)	79.9 ± 17.1
FEV1/FVC ratio (*n* = 285)	71.9 ± 13.2
ppoFEV1	50.56 ± 11.24
Pattern of respiratory function (*n* = 285) Normal Obstructive Restrictive	104 (36.5%) 118 (41.4%) 63 (22.1%)
KCO (% of predicted) (*n* = 132)	78.13 ± 21.07
PAD (mm)	26.0 (24.0–28.0)
NPAD (mm/m^2^)	14.4 ± 2.0 Median 14.0 (13.0–15.6)
Ratio PAD/AoD	0.8 ± 0.1
NAC Yes/No (*n* = 291)	90(30.9%)/201(69.1%)
NAR Yes/No (*n* = 291)	4 (1.4%)/287 (98.6%)
NSCLC/other malignancies	289 (98.3%)/5 (1.7%)
SqCLC/malignancies other than SqCLC	146 (49.7%)/148 (50.3%)
Pathologic stage I/II/IIIA/IIIB/IV (*n* = 282/289 NSCLC)	21 (7.4%)/71 (25.2%)/176 (62.4%)/14 (5%)

BMI: Body Mass Index; BSA: Body Surface Are; CCI Charlson Comorbidity Index; FEV1: Forced Expiratory Volume in 1 s; FVC: Forced Vital Capacity; ppoFEV1: postoperative Forced Expiratory Volume in 1 s; KCO: Coefficient of the lung for carbon monoxide; PAD: Pulmonary Artery Diameter, NPAD: Normalized Pulmonary Artery Diameter; AoD: Aorta Diameter; NAC: Neoadjuvant Chemotherapy; NAR: Neoadjuvant Radiotherapy; NSCLC: Non-Small Cell Lung Cancer; SqCLC: Squamous Cells Lung Cancer.

**Table 2 cancers-12-01515-t002:** Correlations between NPAD and PAD/aorta diameter (AoD) ratio with morphometric, clinical and laboratory variables.

Feature	Normalized Pulmonary Artery Diameter		*p*	Ratio PAD/AoD	*p*
	<Median Value	≥Median Value			
Mean age	61.48 ± 9.89	64.70 ± 9.74	0.0049	r = −0.186	0.0015
Men	113 (53.6%)	98 (46.4%)	0.052	0.79 ± 0.11	0.31
Women	34 (41.0%)	49 (59.0%)		0.81 ± 0.11	
Smoke (*n* = 291) Current Never Smokers	131 (49.1%) 15 (62.5%)	136 (50.9%) 9 (37.5%)	0.21	0.79 ± 0.11 0.80 ± 0.09	0.71
Mean cumulative tobacco consumption (Pack/Years) (*n* = 291)	39.53 ± 24.84	41.41 ± 22.13	0.39	r = −0.065	0.26
Right side	59 (43.7%)	76 (56.3%)	0.047	0.81 ± 0.11	0.067
Left side	88 (55.4%)	71 (44.6%)		0.78 ± 0.11	
Weight (kg)	76.04 ± 13.96	67.65 ± 13.76	0.00000021	r = 0.019	0.74
Height (cm)	171.4 ± 8.2	167.3 ± 7.5	0.000081	r = 0.011	0.85
BMI (kg/m^2^)	25.87 ± 4.26	24.09 ± 4.13	0.00024	r = 0.032	0.59
Underweight	7 (38.9%)	11 (61.1%)	0.78	0.78 ± 0.11	0.77
Normal weight	58 (42.3%)	79 (57.7%)		0.79 ± 0.11	
Overweight	59 (57.8%)	43 (42.2%)		0.80 ± 0.11	
Obesity	23 (62.2%)	14 (37.8%)		0.79 ± 0.11	
BSA (m^2^)	1.88 ± 0.19	1.76 ± 0.19	<0.0000001	r = 0.012	0.84
CCI	4.89 ± 1.64	5.34 ± 1.83	0.041	r = −0.120	0.040
Baseline dyspnea—Modified Borg Dyspnea Scale ≥2 (*n* = 288) Yes No	28 (52.8%) 114 (48.5%)	25 (47.2%) 121 (51.5%)	0.57	0.80 ± 0.09 0.80 ± 0.11	0.56
Mean FEV1 (% of predicted)	79.84 ± 16.02	79.90 ± 18.10	0.97	r = −0.040	0.50
Mean FEV1/FVC (*n* = 285)	72.31 ± 12.79	71.55 ± 13.68	0.81	r = 0.022	0.72
Pattern of respiratory function (*n* = 285) Obstructive Normal Restrictive	60 (50.8%) 51 (49.0%) 31 (49.2%)	58 (49.2%) 53 (51.0%) 32 (50.8%)	0.83	0.79 ± 0.12 0.79 ± 0.11 0.80 ± 0.11	0.54
Mean ppoFEV1	50.68 ± 11.20	50.43 ± 11.29	0.88	r = −0.027	0.69
NAC (*n* = 291) Yes No	46 (51.1%) 100 (49.75%)	44 (48.9%) 101 (50.25%)	0.83	0.80 ± 0.11 0.79 ± 0.11	0.73
NAR (*n* = 291) Yes No	2 (50%) 144 (50.17%)	2 (50%) 143 (49.83%)	0.62	0.83 ± 0.04 0.79 ± 0.11	0.48
Type of malignancy NSCLC Other malignancies	142 (49.1%) 5 (100%)	147 (50.9%) 0 (0%)	0.07	0.80 ± 0.11 0.78 ± 0.11	0.76
SqCLC	72 (49.3%)	74 (50.7%)	0.82	0.80 ± 0.12	0.28
malignancies other than SqCLC	75 (50.7%)	73 (49.3%)		0.80 ± 0.11	
Pathologic stage (*n* = 282/289 NSCLC) I–II III–IV	46 (50.0%) 92 (48.4%)	46 (50.0%) 98 (51.6%)	0.80	0.79 ± 0.11 0.80 ± 0.11	0.44

BMI: Body Mass Index; BSA: Body Surface Are; CCI Charlson Comorbidity Index; FEV1: Forced Expiratory Volume in 1 s; FVC: Forced Vital Capacity; KCO: Coefficient of the lung for carbon monoxide; ppoFEV1: postoperative Forced Expiratory Volume in 1 s; NAC: Neoadjuvant Chemotherapy; NAR: Neoadjuvant Radiotherapy; NSCLC: Non-Small Cell Lung Cancer; SqCLC: Squamous cells lung cancer; PAD: Pulmonary Artery Diameter, NPAD: Normalized Pulmonary Artery Diameter; AoD: Aorta Diameter.

**Table 3 cancers-12-01515-t003:** Correlations between patient’s and tumor related factors with occurrence of respiratory failure requiring intubation and ARDS.

Feature	Need of Mechanical Ventilation			ARDS		
	Yes	No	*p*	Yes	No	*p*
Mean age	66.26 ± 9.53	62.50 ± 9.91	0.028	67.43 ± 8.78	62.60 ± 9.95	0.020
Men	38 (18.0%)	173 (82.0%)	0.075	24 (11.4%)	187 (88.6%)	0.29
Women	8 (9.64%)	75 (90.36%)		6 (7.2%)	77 (92.8%)	
Smoke (*n* = 291) Current Never Smokers	43 (16.1%) 3 (12.5%)	224 (83.9%) 21 (87.5%)	0.86	28 (10.5%) 2 (8.3%)	239 (89.5%) 22 (91.7%)	0.99
Mean cumulative tobacco consumption (Pack/Years) (*n* = 291)	41.96 ± 21.80	40.19 ± 23.85	0.67	43.53 ± 22.22	40.12 ± 23.67	0.41
Right side	29 (21.5%)	106 (78.5%)	0.011	19 (14.1%)	116 (85.9%)	0.043
Left side	17 (10.7%)	142 (89.3%)		11 (6.9%)	148 (93.1%)	
Weight (Kg)	71.22 ± 14.94	71.96 ± 14.39	0.73	70.27 ± 14.18	72.02 ± 14.51	0.53
Height (cm)	170.61 ± 8.80	169.1 ± 8.0	0.28	168.1 ± 6.6	169.4 ± 8.3	0.39
BMI (kg/m^2^)	24.45 ± 4.64	25.08 ± 4.21	0.39	24.86 ± 4.76	24.99 ± 4.23	0.87
BSA (m^2^)	1.82 ± 0.20	1.817 ± 0.196	0.92	1.79 ± 0.18	1.821 ± 0.199	0.44
CCI	6.20 ± 1.92	4.90 ± 1.64	0.00032	5.93 ± 1.57	5.02 ± 1.75	0.0043
Baseline dyspnea—Modified Borg Dyspnea Scale >2 (*n*= 288) Yes No	8 (15.1%) 36 (15.3%)	45 (84.9%) 199 (84.7%)	0.97	5 (9.4%) 24 (10.2%)	48 (9.6%) 211 (89.8%)	0.86
Mean FEV1 (% of predicted)	76.22 ± 17.11	80.55 ± 17.00	0.18	78.53 ± 14.78	80.02 ± 17.33	0.64
Mean FEV1/FVC (*n* = 285)	73.41 ± 15.22	71.66 ± 12.84	0.58	75.29 ± 13.02	71.56 ± 13.22	0.24
Pattern of respiratory function (*n* = 285) Obstructive Normal Restrictive	18 (15.3%) 12 (11.5%) 14 (22.2%)	100 (84.7%) 92 (88.5%) 49 (77.8%)	0.18	10 (8.5%) 8 (7.7%) 10 (15.9%)	108 (91.5%) 96 (92.3%) 53 (84.1%)	0.098
Mean KCO (*n* = 132) (% of predicted)	70.25 ± 19.32	79.88 ± 21.05	0.041	65.43 ± 15.00	79.64 ± 21.18	0.016
Mean ppoFEV1	47.83 ± 9.81	51.07 ± 11.42	0.12	49.08 ± 10.70	50.73 ± 11.29	0.54
NAC (*n* = 291) Yes No	10 (11.1%) 36 (17.9%)	80 (88.9%) 165 (82.1%)	0.14	5 (5.6%) 25 (12.4%)	85 (94.4%) 176 (87.6%)	0.074
NAR (*n* = 291) Yes No	0 (0.0%) 46 (16.0%)	4 (100.0%) 241 (84.0%)	0.86	0 (0.0%) 30 (10.4%)	4 (100.0%) 257 (89.6%)	0.88
Type of malignancy NSCLC Other malignancies	46 (15.9%) 0 (0.0%)	243 (84.1%) 5 (100.0%)	0.73	30 (10.4%) 0 (0.0%)	259 (89.6%) 5 (100.0%)	0.99
SqCLC	25 (17.1%)	121 (82.9%)	0.49	15 (10.3%)	131 (89.7%)	0.97
Malignancies other than SqCLC	21 (14.2%)	127 (85.8%)		15 (10.1%)	133 (89.9%)	
Pathologic stage (*n* = 282/289 NSCLC) I–II III–IV	18 (19.6%) 28 (14.7%)	74 (80.4%) 162 (85.3%)	0.30	12 (13.4%) 18 (9.47%)	80 (89.9%) 172 (90.5%)	0.36
PAD (mm)	26.74 ± 3.70	25.80 ± 3.34	0.22	26.37 ± 3.37	25.90 ± 3.42	0.60
NPAD (≥median) Yes No	31 (21.1%) 15 (10.2%)	116 (78.9%) 132 (89.8%)	0.010	21 (14.3%) 9 (6.1%)	126 (85.7%) 138 (93.9%)	0.021
PAD/AoD ratio	0.82 ± 0.13	0.79 ± 0.11	0.18	0.80 ± 0.12	0.79 ± 0.11	0.85

BMI: Body Mass Index; BSA: Body Surface Are; CCI Charlson Comorbidity Index; FEV1: Forced Expiratory Volume in 1 s; FVC: Forced Vital Capacity; KCO: Coefficient of the lung for carbon monoxide; ppoFEV1: postoperative Forced Expiratory Volume in 1 s; NAC: Neoadjuvant Chemotherapy; NAR: Neoadjuvant Radiotherapy; NSCLC: Non-Small Cell Lung Cancer; SqCLC: Squamous cells lung cancer; PAD: Pulmonary Artery Diameter, NPAD: Normalized Pulmonary Artery Diameter; AoD: Aorta Diameter.

**Table 4 cancers-12-01515-t004:** Correlations between patient’s and tumor related factors with postoperative mortality.

Feature	30-Day Mortality		
	Dead	Alive	*p*
Mean age	69.76 ± 7.47	62.47 ± 9.92	0.00046
Men	18 (8.5%)	193 (91.5%)	0.98
Women	7 (91.6%)	76 (8.4%)	
Smoke (*n* = 291) Current Never Smokers	22 (8.3%) 3 (12.5%)	245 (91.7%) 21 (87.5%)	0.74
Mean cumulative tobacco consumption (Pack/Years) (*n* = 291)	41.08 ± 23.73	40.41 ± 23.53	0.77
Right side	18 (13.3%)	117 (86.7%)	0.0062
Left side	7 (4.4%)	152 (95.6%)	
Weight (kg)	71.92 ± 15.45	71.84 ± 14.39	0.93
Height (cm)	167.70 ± 7.10	169.50 ± 8.20	0.31
BMI (kg/m^2^)	25.53 ± 4.98	24.93 ± 4.21	0.51
BSA (m^2^)	1.80 ± 0.20	1.80 ± 0.19	0.73
CCI	6.41 ± 1.81	4.98 ± 1.69	0.000086
Baseline dyspnea—Modified Borg Dyspnea Scale >2 (*n* = 288) Yes No	4 (7.6%) 21 (8.9%)	49 (92.4%) 214 (91.1%)	0.99
Mean FEV1 (% of predicted)	78.24 ± 14.78	80.02 ± 17.28	0.65
Mean FEV1/FVC (*n* = 285)	72.13 ± 15.16	71.91 ± 13.07	0.60
Pattern of respiratory function (*n* = 285) Obstructive Normal Restrictive	12 (10.2%) 3 (2.9%) 8 (12.7%)	106 (89.8%) 101 (97.1%) 55 (87.3%)	0.042
Mean KCO (*n* = 132) (% of predicted)	73.09 ± 22.01	78.59 ± 20.93	0.42
Mean ppoFEV1	45.86 ± 8.55	51.06 ± 11.38	0.047
NAR (291) Yes No	0 (0.0%) 25 (89.7%)	4 (100.0%) 262 (91.3%)	0.78
NAC (291) Yes No	4 (4.4%) 21 (10.5%)	86 (95.6%) 180 (89.5%)	0.091
Type of malignancy NSCLC Other malignancies	25 (8.5%) 0 (0.0%)	264 (91.5%) 5 (100.0%)	0.90
SqCLC	12 (8.2%)	134 (91.8%)	0.86
Malignancies other than SqCLC	13 (8.8%)	135 (91.2%)	
Pathologic stage (*n* = 282/289 NSCLC) I–II III–IV	11 (12.0%) 14 (7.4%)	81 (88.0%) 176 (92.6%)	0.20
PAD (mm)	26.60 ± 3.60	25.88 ± 3.39	0.38
NPAD (≥median) Yes No	8 (5.4%) 17 (11.6%)	139(94.6%) 130(88.4%)	0.06
PAD/AoD ratio	0.808 ± 0.141	0.793 ± 0.107	0.67

BMI: Body Mass Index; BSA: Body Surface Are; CCI Charlson Comorbidity Index; FEV1: Forced Expiratory Volume in 1 s; FVC: Forced Vital Capacity; KCO: Coefficient of the lung for carbon monoxide; ppoFEV1: postoperative Forced Expiratory Volume in 1 s; NAC: Neoadjuvant Chemotherapy; NAR: Neoadjuvant Radiotherapy; NSCLC: Non-Small Cell Lung Cancer; SqCLC: Squamous cells lung cancer; PAD: Pulmonary Artery Diameter, NPAD: Normalized Pulmonary Artery Diameter; AoD: Aorta Diameter.

**Table 5 cancers-12-01515-t005:** Multivariable analysis.

Feature		OR	IC 95%	*p*
**IMV**	CCI > 5	3.8	1.76–822	0.0009
	Right pneumonectomy	2.37	1.20–4.71	0.013
	NPAD > 14 mm/m^2^	2.16	1.08–4.33	0.029
**ARDS**	CCI > 5	2.55	1.17–5.59	0.018
	NPAD > 14 mm/m^2^	2.52	1.10–5.77	0.028
**DEATH**	CCI > 5	5.56	1.99–15.54	0.0011
	Right pneumonectomy	4.11	1.46–11.56	0.0074
	NPAD > 14 mm/m^2^	3.39	1.15–9.95	0.026

IMV: Invasive Mechanical Ventilation, CCI Charlson Comorbidity Index; NPAD: Normalized Pulmonary Artery Diameter; ARDS: Acute Respiratory Distress Syndrome.

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
