# Peer review of "Normalized Pulmonary Artery Diameter Predicts Occurrence of Postpneumonectomy Respiratory Failure, ARDS, and Mortality"

_cancers, 2020, doi:10.3390/cancers12061515_

Round 1

Reviewer 1 Report

The authors report an interesting paper regarding the risk factors for ARDS and mortality after pneumonectomy. This is an update on a large series of pneumonectomy of a previous pilot study.

The paper is well written and in my opinion is very interesting, however before eventual publication it deserve some changes:

  1. The Materials and methods section should follow the introduction section and should precede the results.
  2. The tables are well documented however some abbreviations are not reported in the text and should be specified as legend at the end of the table.
  3. The introduction and discussion sections are too long and should be shortened.
  4. The results of multivariable analysis should be reported as table or included in the same tables of univariate analysis.
  5. The discussion is interesting, however the authors should comment about the idea to include more specific preoperative exams (e.g. right catheterization in those patients with high NPAD) to better evaluate for example the presence of pulmonary hypertension.

Author Response

Rev 1: The authors report an interesting paper regarding the risk factors for ARDS and mortality after pneumonectomy. This is an update on a large series of pneumonectomy of a previous pilot study.

Au: We would like to thank the rewiever for kind appreciation

Rev 1: The paper is well written and in my opinion is very interesting, however before eventual publication it deserve some changes:

Au: Thank Again

Rev 1 : The Materials and methods section should follow the introduction section and should precede the results.

Au: We followed the instructions of  Cancers asking that The Materials and methods section should follow the discussion section

Rev 1: The tables are well documented however some abbreviations are not reported in the text and should be specified as legend at the end of the table.

Au: We apologize for the omissions in the definition of abbreviations. We have corrected the tables and report in the text and the tables the missing abbreviations.

Rev 1: The introduction and discussion sections are too long and should be shortened.

Au: As you have suggested, we removed some redundant paragraph in the  introduction and discussion.

Rev 1: The results of multivariable analysis should be reported as table or included in the same tables of univariate analysis.

Au: As you have suggested, we have reported results of multivariable analysis in a new table (Table 5 in the current version).

Rev 1: The discussion is interesting, however the authors should comment about the idea to include more specific preoperative exams (e.g. right catheterization in those patients with high NPAD) to better evaluate for example the presence of pulmonary hypertension.

Au: As you have suggested, we have added comment on this subject. In our center, right catherization is not carried out routinely, but discussed on a case-by-case basis in case of patent pulmonary hypertension at echography. On the basis of our results, the idea of performing this invasive assessment of pulmonary pressure in patients with high NPAD (which otherwise reveals moderate increase in pulmonary pressure) should probably be tested in the setting of prospective studies. We added this concept in the discussion.

We would like to thank the Reviewer for the help in improving our paper. Hopefully, it is now suitable for publication.

Reviewer 2 Report

Your manuscript is well written, with the employed workflow well described. The statistical analysis have been performed on an adequate sample size. Authors should clarify why they choosed to use one different model for the multivariate analysis to include KCO evaluation.  The results are interesting, however authors should better discuss  the reasons why NPAD in patients with PAD/AoD > 1 and affected by tumors different than NSCLC is not a predictor for post-operative reintubation.

Author Response

Rev 2: Your manuscript is well written, with the employed workflow well described. The statistical analysis have been performed on an adequate sample size.

Au: We would like to thank the rewiever for kind appreciation

Rev 2: Authors should clarify why they choosed to use one different model for the multivariate analysis to include KCO evaluation. 

Au: We agree with the reviewer about possible misunderstanding. KCO measurement was not available for all patients, as showed in both text and tables of the initial manuscript. Due to this lack of data, we used  models without KCO (allowing to study the entire population) and models with KCO (which obviously include only patients undergoing this test). For better clarity, in agreement with Reviewer 1 request, we showed in a separate table results of multivariate analysis.

Rev 2: The results are interesting, however authors should better discuss the reasons why NPAD in patients with PAD/AoD > 1 and affected by tumors different than NSCLC is not a predictor for post-operative reintubation.

Au: We apologize for the misunderstanding due to confusion generated by the abbreviation choice. SCLC does no correspond to small cell lung cancer but corresponds to squamous cells lung cancer. To avoid confusion we changed the abbreviation SCLC for SqCLC: almost all the patients in our study had non-small cell lung cancer. Independently from histologic type, higher NPAD and  PAD/AoD > 1 identify different conditions, i.e. mild or patent pulmonary hypertension, as underlined in the discussion section of the paper.

We would like to thank the Reviewer for the help in improving our paper. Hopefully, it is now suitable for publication.

Reviewer 3 Report

Dear authors,

thank you for giving me the opportunity to review your article. This an impressive work with a large series of patients undergoing pneumonectomy in a high volume center. I have some comments:

  • How did you define under-, overweight? not clear for me..
  • You should add the definition of the abreviations. (CCI, NAC.... on the table for example..)
  • I am surprised by the high rate of pneumonectomy for SCLC, what is your indication? These patients should have previously radio and chemotherapy which can complicate post-operative course...
  • More generally, did you also analyze patients with pre-op chemo and/or radiotherapy? this point could influence post-op outcomes as well.

What is your limit in term of pulmonary arterial pressure to propose a pneumonectomy? Did all patient have an echography or a right side catheter to control the pressure?

Author Response

Rev 3: Dear authors, thank you for giving me the opportunity to review your article. This an impressive work with a large series of patients undergoing pneumonectomy in a high volume center. I have some comments:

Au: We would like to thank the rewiever for kind appreciation

Rev 3: How did you define under-, overweight? not clear for me..

Au: We have corrected the missing definitions and reported in the tables the definitions of under and overweight.

Rev 3: You should add the definition of the abbreviations. (CCI, NAC.... on the table for example..)

Au: We apologize for the omissions in the definition of abbreviations. We have corrected the tables and report in the text and in the tables the missing abbreviations.

Rev 3 : I am surprised by the high rate of pneumonectomy for SCLC, what is your indication?

Au: We apologize for the confusion due to the abbreviation choice. SCLC does no correspond to small cell lung cancer but corresponds to squamous cells lung cancer. So we changed the abbreviation for SqCLC to avoid confusion.

Rev3: These patients should have previously radio and chemotherapy which can complicate post-operative course... More generally, did you also analyze patients with pre-op chemo and/or radiotherapy? this point could influence post-op outcomes as well

Au: We have effectively analysed patients wit pre-op chemo and/or radiotherapy defined as NAC and NAR. This did not influence the post-op outcomes. Again we apologize for the missing abbreviations in the tables.

Rev 3: What is your limit in term of pulmonary arterial pressure to propose a pneumonectomy? Did all patient have an echography or a right side catheter to control the pressure?

Au: All patients undergoing pneumonectomy have an echography in our center, and right catherization is not carried out in routine basis. It Is discussed on a case-by-case basis in patients with patent pulmonary hypertension at cardiac echography.

We would like to thank the Reviewer for the help in improving our paper. Hopefully, it is now suitable for publication.

Round 2

Reviewer 3 Report

Dear authors, thank you for your responses. The manuscript is appropriate for publication.